# High-Gain Millimeter-Wave Patch Array Antenna for Unmanned Aerial Vehicle Application

**DOI:** 10.3390/s21113914

**Published:** 2021-06-06

**Authors:** Kyei Anim, Jung-Nam Lee, Young-Bae Jung

**Affiliations:** 1Electronics Engineering Department, Hanbat National University, Daejeon 34158, Korea; kyeianim@gmail.com; 2Electronics and Telecommunications Research Institute (ETRI), Daejeon 34129, Korea; jnlee77@etri.re.kr

**Keywords:** aperture-coupled feed, cavity backed patch, gain, lightweight, millimeter-wave, stripline, substrate integrated waveguide, surface wave, unmanned aerial vehicle

## Abstract

A high-gain millimeter-wave patch array antenna is presented for unmanned aerial vehicles (UAVs). For the large-scale patch array antenna, microstrip lines and higher-mode surface wave radiations contribute enormously to the antenna loss, especially at the millimeter-wave band. Here, the element of a large patch array antenna is implemented with a substrate integrated waveguide (SIW) cavity-backed patch fed by the aperture-coupled feeding (ACF) structure. However, in this case, a large coupling aperture is used to create strongly bound waves, which maximizes the coupling level between the patch and the feedline. This approach helps to improve antenna gain, but at the same time leads to a significant level of back radiation due to the microstrip feedline and unwanted surface-wave radiation, especially for the large patch arrays. Using the SIW cavity-backed patch and stripline feedline of the ACF in the element design, therefore, provides a solution to this problem. Thus, a full-corporate feed 32 × 32 array antenna achieves realized gain of 30.71–32.8 dBi with radiation efficiency above 52% within the operational band of 25.43–26.91 GHz. The fabricated antenna also retains being lightweight, which is desirable for UAVs, because it has no metal plate at the backside to support the antenna.

## 1. Introduction

Unmanned aerial vehicles (UAVs) and drone technologies have seen a rapid growth of interest due to the breakthroughs in microprocessors and artificial intelligence (AI) [1,2] to make them more accessible and affordable. UAVs can operate either autonomously or remotely controlled [3,4] to carry out tasks such as search and rescue operations, security patrol, cargo transportation, and provide seamless wireless coverage where no communication infrastructure is available [5,6,7].

UAV antennas must, therefore, achieve a good trade-off between the commercial criteria and technical design issues. This includes high performances (i.e., gain), low power consumption, restrictive physical properties (e.g., size and weight), and low cost [8]. Owing to the trend of miniaturized electronic devices and increasing demands for high data rates, microstrip patch antennas (MPAs) are better suited for UAV technologies than other antenna types, especially at the millimeter-wave (mm-wave) band. This is because, at mm-wave frequencies, the MPAs become small and lightweight. However, the standard MPAs having huge radiation aperture contradict the high-gain requirement for the UAV applications due to the lower radiation efficiency. This is because of the significant losses attributable to higher-mode surface wave radiation, leaky-wave radiation from the feeding network, and ohmic loss by the dielectric substrate, especially in higher frequencies. Thus, a large-scale array of standard MPA cannot achieve superior gain results in the millimeter-wave band. A more innovative design must be investigated.

Several high-gain horn antennas have been reported in the literature with promising electrical and physical features [9,10,11]. For example, they have a compact size for flush-mounted installation on the UAV platform to improve aerodynamics and broadband for increased capacity. The horn antenna gain ranges up to 25 dBi, with 10~20 dBi being typical. Nonetheless, some UAV applications demand extremely high antenna gain above 25 dBi in order to have a reliable communication link with the ground control stations in higher frequency bands. This is because the path losses increase with the increase of operating frequency and distance. In such a situation, the horn antennas cannot be chosen over the patch array antenna with large radiation aperture.

In this paper, a corporate-fed 32 × 32 patch array antenna is developed at the millimeter-wave band for UAV applications. To achieve higher gain, the element’s aperture-coupled feed (ACF) is modified by increasing the coupling slot size. This approach increases the coupling levels between the patch and the feedline to improve radiation efficiency. However, at the same time, strong surface waves and back radiation due to the feedline occur, especially in the large-scale patch arrays in higher frequency bands. This phenomenon causes significant antenna losses to undermine the gain improvement. Hence, the element antenna is implemented with SIW cavity-backed patch to create strongly bound input power and suppress any undesired higher-mode surface wave. Also, a stripline feedline is used in the ACF scheme to minimize back radiation level. The experimental results indicated a high gain of 30.71–32.8 dBi with the radiation efficiency of at least 52% within operational band of 25.43–26.91 GHz. The fabricated antenna retains low weight because it has no metal plate to back the antenna’s bottom ground plane. Thus, the external WR-28 waveguide-to-coax adapter is mounted directly at the antenna’s backside for measurement.

## 2. Radiating Element Design

The geometry of the proposed array element is shown in Figure 1, which is a multilayer structure with three dielectric substrates made of Taconic *TLY-5* (i.e., Substrate #1, Substrate #2, and Substrate #3, *ε_r_* = 2.2, *tanδ* = 0.0009, height *h* = 0.508 mm). It also consists of three copper layers, i.e., top layer GND #1, middle or isolating layer GND #2, and bottom layer GND #3. The radiating patch resides on substrate 1, as shown in Figure 1a. Several metal vias spaced along the rectangular opening are installed in the substrate and connected to the top and middle ground planes (GND #1 and GND #2), thus forming the SIW cavity that is the backing of the patch. The metal vias’ diameter *d_v_*, and spacing *s_v_* are carefully optimized to reduce the leakage loss and provide feasible fabrication. The rectangular patch on the top surface of substrate #1 is fed by a stripline feedline integrated on substrate #2 and substrate #3 through the coupling slot or aperture in the isolating ground plane GND #2, thus constituting the ACF scheme (see Figure 1a). The coupling slot is centrally positioned under the patch to produce lower cross-polarization due to the symmetry of the configuration [9] according to the *X*- and *Y*-axis, as illustrated in Figure 1b.

The implemented feeding configuration in this element design is similar to the conventional ACF. The only difference is at the coupling-slot size. In contrast to the conventional ACF that contains a small non-resonant coupling slot, the proposed element antenna has a relatively large coupling aperture whose width *w_s_* is about seven times larger than that of the conventional coupling slot. This modification shows a significant improvement of the coupling levels (see Figure 2a) as the magnetic polarization of the slot (a dominant mechanism for the coupling) is highly dependent on the size and shape of the aperture or slot [12]. Although maximum coupling between the feedline and the patch is obtained in Figure 2a with larger coupling-aperture size, a smaller non-resonant aperture produces lower back radiation with a better impedance matching to result in less spurious radiation and better efficiency. Thus, a large coupling aperture utilized in this element antenna causes significant spurious radiation due to the strong higher-mode surface wave and transmission line radiations to lower efficiency, especially in large-scale patch array in higher frequency bands. Also, the microstrip line of the ACF underneath the patch causes undesired reactive loading effects at the element antenna’s input port because of the distortion in the electric field between the isolating ground plane and the patch [13].

To overcome these drawbacks of the ACF with large aperture size, SIW cavity-backed patch has been combined with a stripline feeding line to implement the radiating structure, as shown in Figure 1. This configuration significantly suppresses undesired higher-mode surface waves, as illustrated in Figure 2b, as well as back radiation to maximize the radiation efficiency. Figure 3a indicates that increasing the slot width *w_s_* leads to a corresponding increase in radiation efficiency. This validates this design concept since the radiation efficiency peaks at *w_s_* = 0.146λ, which is better than conventional ACF. Hence, the element antenna exhibits gain variation less than 1 dBi from 7.82 to 6.89 dBi within the band of interest, as shown in Figure 3b. A peak gain of 7.82 dB occurs at 26.4 GHz with a radiation efficiency of more than 89%. See Figure 4a, the simulated model of element antenna operates from 25.58 to 27.04 GHz for |S11| < −10 dB (5.2%). The simulation results in Figure 4b indicates half power beamwidth (HPBW) of 70° and 73° in the XOZ- and YOZ-plane, respectively. The level of cross-polarization in the boresight direction is less than −20 dB. The geometric dimensions of the radiating element can be found in Table 1.

The geometry of the element antenna in Figure 1, however, can lead to power leakage to the parallel plate mode due to the use of the stripline feed to cause a decrease in radiation efficiency. Figure 5 shows the parallel plate mode of the stripline, indicating the electric field lines due to the voltage |Vs| applied to the coupling slot. Thus, the amount of parallel plate mode is contingent on the amplitude of |Vs|. The impedance seen looking from the slot toward the patch is purely real (*Ra*) at resonance for the aperture-coupled antenna (ACA), as illustrated in the circuit model in Figure 6a [14]. As seen from Figure 6b, *Ra* is very low compared to the 60-Ω input impedance of the element antenna within the operating frequency band. The patch, therefore, appears to behave like a circuit that shorts out the slot at these resonant frequencies, resulting in the parallel plate mode suppression. It should be noted that *L_f_* and *Cf* represent the feedline, whereas the patch is characterized by *L_p_*, *C_p_*, and *R_p_*.

## 3. Array Antenna Design

The structure of the 32 × 32 array based on the proposed element antenna is shown in Figure 7a. The antenna’s geometry comprises three substrate layers with a corporate feeding network to distribute power among the constituent elements. Its feeding network has Tee-junctions, which are two-way equal power dividers to ensure uniform distribution of power. Some sections of the feeding lines are bent to achieve 180° phase shift between adjacent elements in the *x*-axis for in-phase excitation, as illustrated in Figure 7b. The center-to-center distance between adjacent radiating patches is 7.2 mm in the *x*-direction and 8 mm in the *y*-direction. The entire radiation aperture of the full-corporate feed 32 × 32 array antenna covers an area of 232.6 mm (20λ_0_) × 257 mm (22.5λ_0_) (*x* × *y*). The height of the antenna is 1.664 mm (0.14 λ_0_). The paths of the input signals to the radiating patches are detailed in Figure 5 as Block III→ Block I. 

Block III: Waveguide-to-stripline Transition-The launched signal in the waveguide excites the matching patch of the transition at 26.2 GHz. The matching patch is located in the feed aperture in the bottom ground plane of substrate #3.

Block II: Feed network-The input signals are electromagnetically coupled to the feedline on the top surface of substrate #3 from the matching patch. The signals are then transferred to and evenly distributed in the feeding network. 

Block I: Radiating part-The SIW cavities on substrate #1, comprising coupling apertures and metal vias, are designed to transfer power between the substrate layers. Thus, the uniformly distributed signals in the feeding network are also coupled electromagnetically through the apertures or slots, which are etched in the middle ground plane, to the patches. The signals are subsequently re-radiated by the patches without the influence of the higher-mode surface waves due to the cavity-backed technique.

### 3.1. Block II: Feed Network

As pointed out previously, the feed network (see Figure 7b) adopts a Tee-junction for signal division. It also contains a microstrip bend (i.e., delay line), which follows the Tee-junction to achieve a 180-degree phase difference between the adjacent elements because the delay line length is adjusted to *λg*/2 (where *λg* is the guided wavelength at the center frequency. With reference to Figure 7b, no additional metal vias were installed along the feedlines of the feed network to suppress the surface waves from feed structure and ground planes (i.e., stripline shielding). This is because it will further increase the complexity of the full array structure. The prototyping of the antenna, thus, becomes more difficult to perform, making the antenna more susceptible to fabrication errors, which eventually deteriorate the measured antenna performance. In addition, the metal via itself could also radiate, especially at a higher frequency, as the size is comparable to the operating wavelength to result in radiation losses. It is, therefore, imperative to take into consideration the coupling between the feedlines without the stripline shielding. Figure 8a shows the coupling coefficients between the feedlines in terms of |S12|, |S21| and |S23|, and |S32|. It can be seen that the coupling coefficient is lower than −30 dB over the frequency range of 24.5–27.5 GHz. This suggests that less than 0.1% of the input power contributes to the coupling between feedlines, which is negligible to reduce efficiency. Figure 8a also indicates that a 180-degree phase difference is achieved by the delay line. 

### 3.2. Block III: Waveguide-to-Stripline Transition Structure

The waveguide-to-stripline transition of the array antenna is shown in Figure 7c. This transition consists of a rectangular waveguide, feedline, waveguide-short pattern, metal vias, and matching patch. The top substrate layer (i.e., antenna’s substrate #2) is cladded with copper layer as the upper ground plane to achieve the stripline configuration. Thus, any undesired transmission line radiation is suppressed to improve the radiation efficiency, especially for the large-scale patch array. On top of the lower substrate (i.e., substrate #3) resides the feeding line. The matching patch is etched on the substrate’s backside with a short-terminated waveguide using the metal vias. WR-28 waveguide was used with dimensions of a = 7.112 mm and b = 3.556 mm. 

Referring to Figure 7c, the waveguide’s dominant TE10 mode transforms to the feedline’s quasi-TEM mode through the radiation mode TM01 of the matching patch. Thus, by adjusting the length and width of the matching patch, the desired operating frequency and good impedance matching can be realized to lower insertion loss. Figure 8b shows the simulated reflection |S11| and transition |S21| characteristics of the optimized waveguide-to-stripline transition in a back-to-back configuration. The transition operates from 24.86 to 28.24 GHz for |S11| < −10 dB (12.7%) with an insertion loss |S21| better than −0.6 dB from 25 to 27.95 GHz. The insertion loss is due to the transmission line (loss = 0.012 dB/mm) and the two transition structures (0.1 dB each) for the back-to-back arrangement. Figure 8b also shows the phase response of the waveguide-to-stripline transition. 

## 4. Results and Discussion

The photographs of the fabricated 32 × 32 array antenna are shown in Figure 9. It is observed that the backside of the antenna has no metal plate attached to the bottom ground plane for lightweight purposes, which is desirable for UAVs. Thus, the WR-28 waveguide to coax adapter is directly connected to the antenna’s feed point for measurement. 

Figure 10a,b exhibit the experimental results of |S11|, gain, and radiation efficiency. There is a good agreement between the simulated and measured results. The measured |S11| < −10 dB bandwidth is 5.65%, i.e., from 25.43 to 26.91 GHz, while the measured gain varies between 30.71 and 32.8 dBi within this band. A peak gain of 32.8 dBi is observed at 26.2 GHz. On the other hand, the simulated gain varies between 33.9 and 35 dBi. The overall disparity between the simulated and measured gain is generally due to the insertion loss of 0.3 dB of the coaxial waveguide adapter or transition. Other contributing factors include copper surface roughness, substrate shrinkage, and assembly tolerance, which are difficult to account for during the full-wave analysis. They significantly affect the performances of antennas in millimeter-wave bands, especially in multilayered structures. The radiation efficiency plotted in Figure 10b is above 52% within the band of operation. The simulated and measured far-field radiation patterns at 25.43, 25.9, 26.2, and 26.9 GHz are shown in Figure 11, respectively. Well-maintained beam shapes at these three frequencies can be observed. The measured HPBWs for all the three frequency points are 2° and 2.2° in XOZ- and YOZ-plane, respectively.

## 5. Conclusions

In summary, this paper presents a millimeter-wave array antenna for unmanned aerial vehicles (UAVs). The array element utilizes a large coupling aperture (compared with the conventional ACF slot size) to enhance coupling power between the feeding line and patch. However, with this design concept, the large-scale patch array results in significant transmission lines and higher-mode surface wave radiations to increase antenna loss. The radiating element is therefore implemented with the SIW cavity-backed patch and stripline feeding line of ACF. Thus, a full-corporate feed 32 × 32 array antenna including the transition achieves gain between 30.7 and 32.8 dBi with radiation efficiency above 52% within the band of interest. Also, the fabricated antenna has no metal plate at the backside to maintain its lightweight attribute, essential for UAV applications.

## Figures and Tables

**Figure 1 sensors-21-03914-f001:**
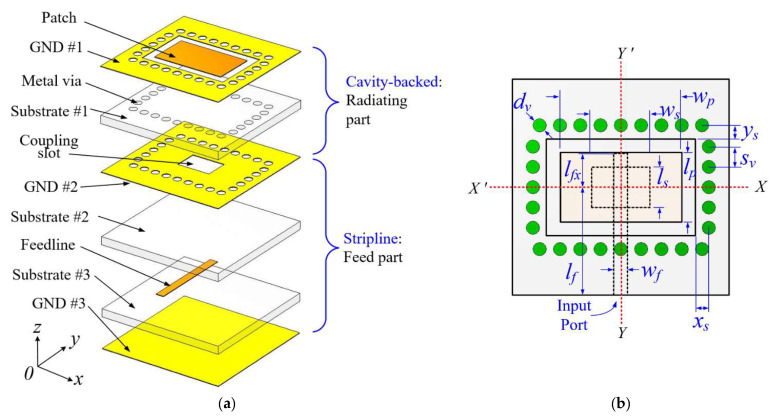
The geometry of element antenna: (**a**) 3D exploded view and (**b**) top view.

**Figure 2 sensors-21-03914-f002:**
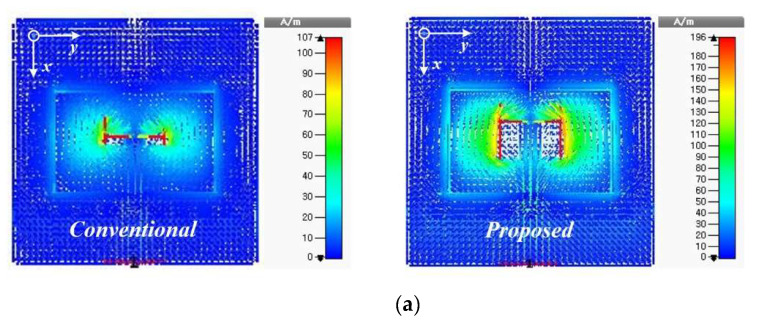
Simulated current distributions on element antenna at 26.2 GHz. (**a**) Conventional versus proposed slot size. (**b**) Conventional versus SIW cavity-backed microstrip patch.

**Figure 3 sensors-21-03914-f003:**
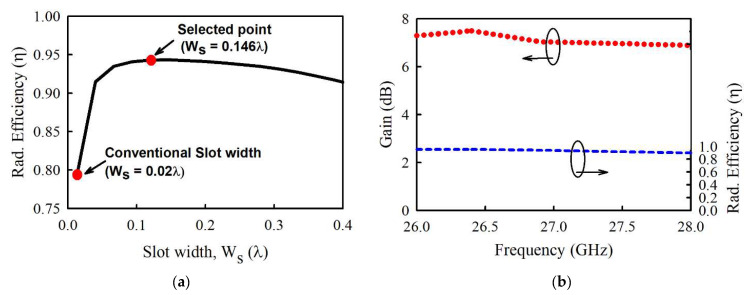
Simulation results of the element antenna: (**a**) Radiation efficiency in variation of coupling slot width, *ws*. (**b**) Gain and radiation efficiency.

**Figure 4 sensors-21-03914-f004:**
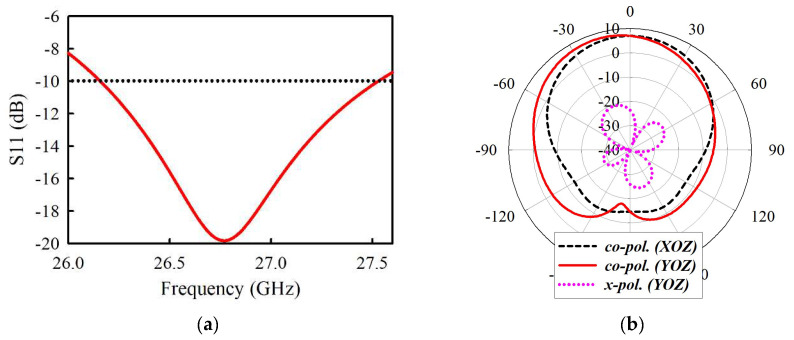
Simulated results of the proposed element antenna. (**a**) |S11|. (**b**) Radiation patterns at 26.2 GHz.

**Figure 5 sensors-21-03914-f005:**
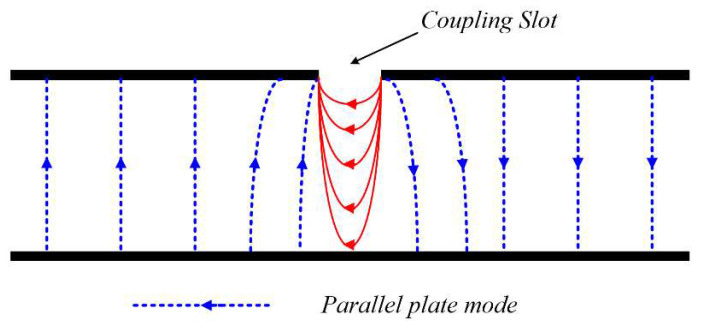
Parallel plate waveguide developed from the ground planes of the stripline of element antenna.

**Figure 6 sensors-21-03914-f006:**
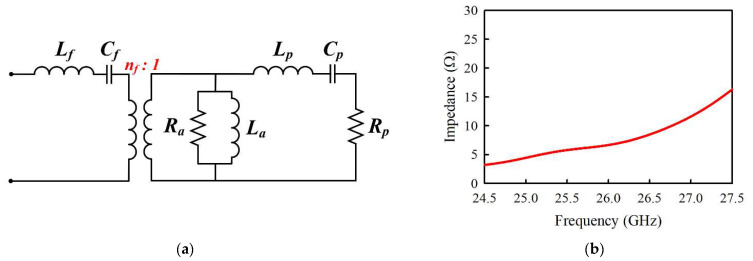
(**a**) Equivalent circuit of aperture-coupled antenna. (**b**) Impedance seen looking from the slot toward the patch within the operating band of the element antenna.

**Figure 7 sensors-21-03914-f007:**
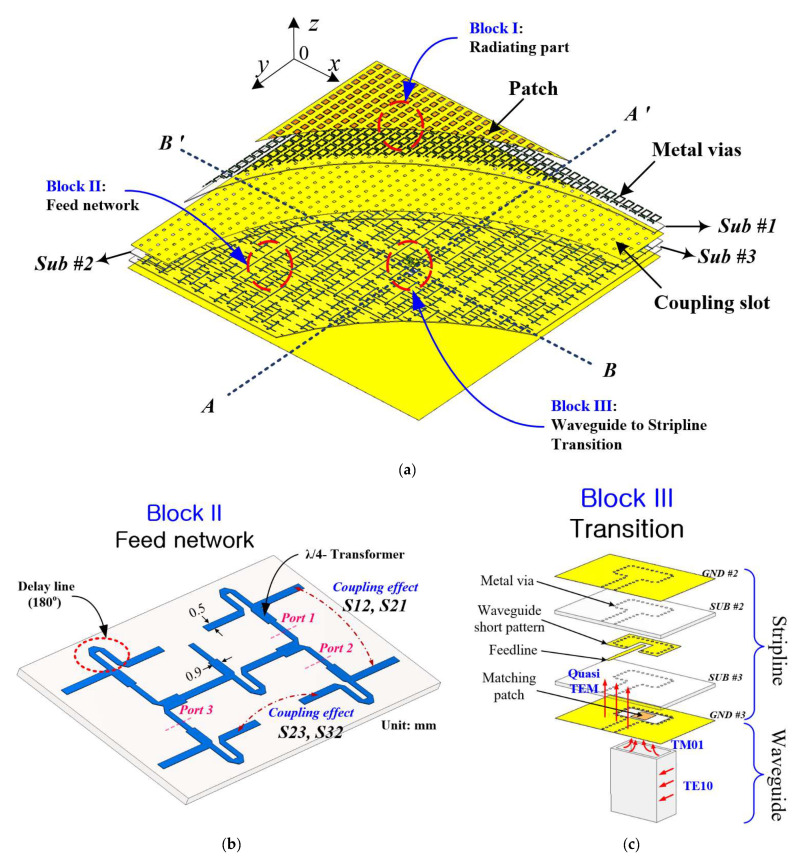
(**a**) Configuration of the proposed array antenna. (**b**) Detailed 3-D view of the corporate-feed network. (**c**) Detailed 3-D view of the waveguide-to-stripline transition.

**Figure 8 sensors-21-03914-f008:**
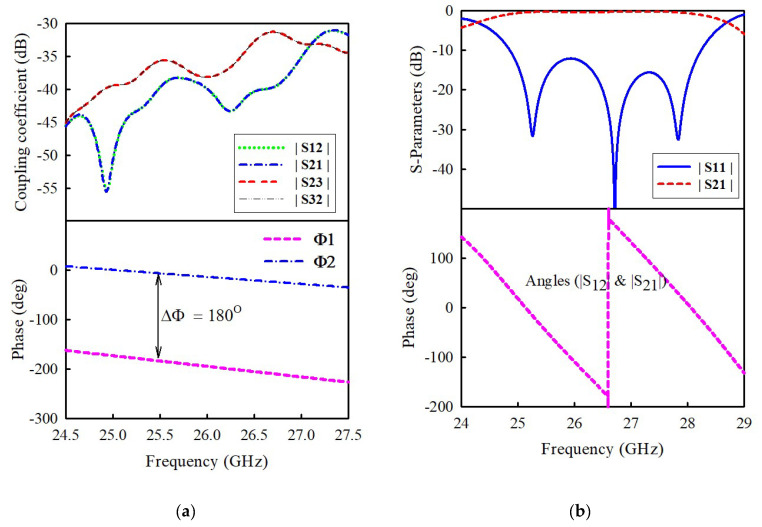
(**a**) Coupling coefficient and phase of feed network. (**b**) Simulated S11, S21, and phase of waveguide-to-stripline transition.

**Figure 9 sensors-21-03914-f009:**
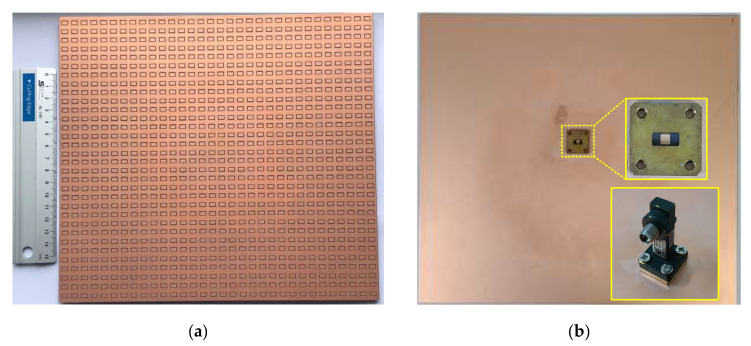
Photograph of the fabricated 32 × 32 array antenna. (**a**) Top view. (**b**) Back view.

**Figure 10 sensors-21-03914-f010:**
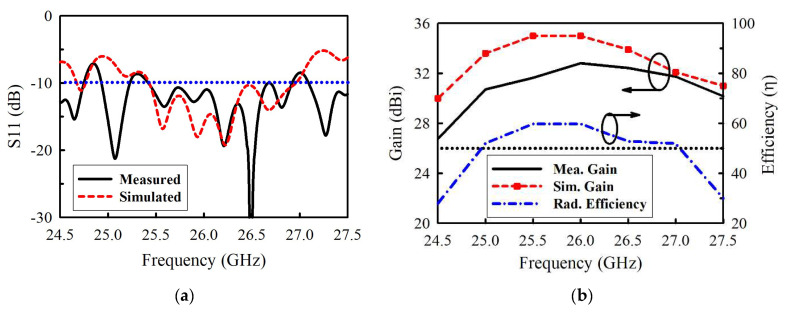
(**a**) Reflection coefficient. (**b**) Gain and efficiency.

**Figure 11 sensors-21-03914-f011:**
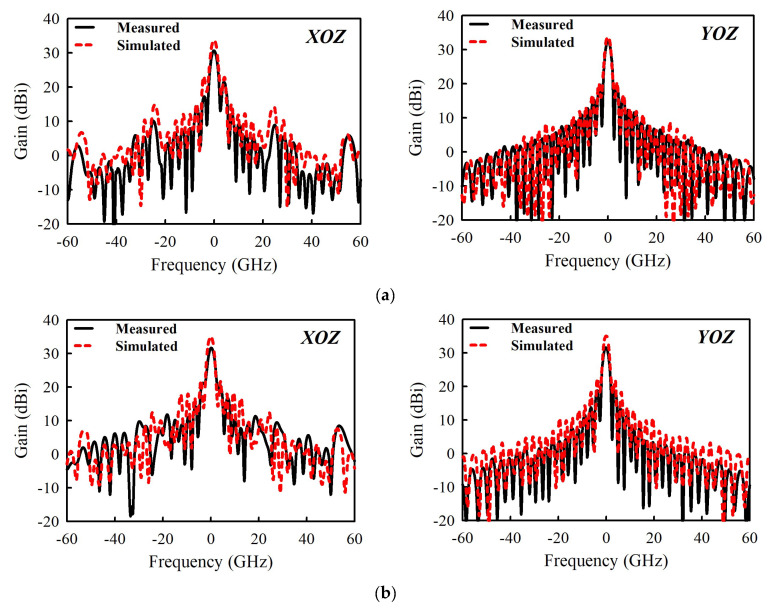
Simulated and measured radiation patterns of the 32 × 32 planar array antenna at different frequency points: (**a**) 25.43, (**b**) 25.9, (**c**) 26.2, and (**d**) 26.9 GHz.

**Table 1 sensors-21-03914-t001:** Dimensions of the Element Patch Antenna.

Parameters	Value	Parameters	Value
*d_v_*	0.50 mm	*w_f_*	0.50 mm
*x_s_*	0.50 mm	*w_p_*	4.50 mm
*y_s_*	0.50 mm	*w_s_*	5.50 mm
*s_v_*	0.75 mm	*l_s_*	3.58 mm
*l_fx_*	1.25 mm	*l_p_*	2.58 mm
*l_f_*	4.00 mm		

## Data Availability

Not applicable.

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
