# Peer review of "High-Gain Millimeter-Wave Patch Array Antenna for Unmanned Aerial Vehicle Application"

_sensors, 2021, doi:10.3390/s21113914_

Round 1

Reviewer 1 Report

Very interesting and comprehensive article dealing w/ mm wave patch antennas. Some small remarks:

  1. It seems to me a counterpart to the patch antenna, for this application, is the aperture horn antenna. It can be flush-mounted in the UAV, has high gain (though not as high as 30 dB as this array), broadband, small dimensions, and high efficiency. Besides that its much simpler than that of this patch array. Authors could comment on that in the introduction.
  2. Fig. 2 presents data in min/max which is very convenient. However, I feel that it has used decibels, if so it's got to be mentioned, alongside the dynamic range.
  3. Table I contains the original template text. The authors should correct it.
  4. Which software was used for the simulation? Which numerical method?
  5. I'd like to know more about fabrication. Was it sent to an external fab facility? Was it built in-home? This complex structure is unfortunately too complex for most researchers, they cannot afford such a precise multi-layer array, so readers would benefit from it.
  6. Fig. 7  should contain a coin, rule, or pen for size reference.

Author Response

Please refer attached files.
I did my best to revise for reviewer's valuable comments.

Sincerely,

Young-Bae Jung

Reviewer 2 Report

The paper presents an interesting solution for a completely shielded patch array antenna with corporate BFN. The radiating element by itself is not particularly original, but the achieved results are indeed worth of notice. However the claimed gain and efficiency performance is not supported by presented data: a radiation efficiency above 52% within the operational band of 25.43 ~ 26.91 GHz is claimed, though Figs. 8b and 9 only show results above 26 GHz. An integration of those graphs is necessary, to support the declared performance.

In addition to that, more details on the corporate BFN should be provided. Fig.5 shows in fact a low-quality image of the whole antenna with BFN. From the picture it appears that no shielding of the strip-line is applied on the BFN. If this is confirmed then detailed comments and at least simulated results should be shown to demonstrate that (a) no parallel-plate mode is excited all along the BFN and (b) coupling between feeding lines is negligible. In my personal opinion both issues are typically present when no shielding is applied.

For the above reasons a major revision is considered to be necessary before this paper could be accepted for publication.

Author Response

(The authors gave the same response as above.)

Round 2

Reviewer 2 Report

Authors have properly integrated the text. The work can now be accepted for publication.